# Analysis of the Effects of Epidural Anesthesia on the Nociception Level Index (NOL^®^) during Abdominal Surgery

**DOI:** 10.3390/jcm13164968

**Published:** 2024-08-22

**Authors:** Alexander Ziebart, David-Jonas Rothgerber, Sophia Woldt, Katharina Mackert, Julia Heiden, Michael Schuster, Jens Kamuf, Eva-Verena Griemert, Robert Ruemmler

**Affiliations:** Department of Anesthesiology, University Medical Centre, Johannes Gutenberg-University Mainz, 55131 Mainz, Germanykatharina.mackert@unimedizin-mainz.de (K.M.);

**Keywords:** epidural anesthesia, Nociception Level Index, nociception monitor, local anesthetic

## Abstract

**Background**: The NOL^®^ system (PMD-200™ Nociception Level Monitor; Medasense Ltd., Ramat Gan, Israel) is used for the real-time detection of physiological nociception in anesthetized patients by assessing the parameters indicative of sympathetic activity, such as photoplethysmography, skin conductance, peripheral temperature, and accelerometry, which are quantified into the NOL^®^-Index. This index is more sensitive than traditional clinical parameters in estimating pain and stress responses. While its effectiveness in general anesthesia is well documented, its efficacy in epidural anesthesia needs further investigation. **Methods**: This retrospective study analyzed NOL^®^-Index dynamics compared to conventional parameters after epidural administration of bupivacaine. Following ethics committee approval, 119 NOL^®^ measurements were retrospectively analyzed after thoracic epidural catheter administration in 40 patients undergoing abdominal and urological surgery. The NOL-Index^®^ was assessed at 0, 1, 3, and 5 min post application and compared to heart rate, blood pressure, and bispectral index dynamics. **Results**: This study showed a significant decrease in the NOL^®^-Index post-local-anesthetic administration with better sensitivity than classical clinical parameters (0 min = 38 ± 11; 1 min = 22 ± 13*; 3 min = 17 ± 11*; 5 min = 12 ± 10*). Higher doses of local anesthetics led to a significant, dose-dependent decrease in NOL^®^-Index (low dose, 5 min = 15 ± 10*; high dose, 5 min = 8 ± 8*). **Conclusions**: This study is the first to demonstrate the effectiveness of the NOL^®^-Index in measuring nociceptive effects following epidural administration, highlighting its potential superiority over conventional parameters and its sensitivity to dose variations.

## 1. Introduction

The main objective of anesthesia is to guarantee patient safety and comfort while improving surgical conditions. When general anesthesia is used for this purpose, it is crucial to have the following three critical components converge: analgesia, hypnosis, and, if necessary, neuromuscular blockade. This convergence is necessary to achieve the overall goal [1,2].

As a result, diverse methodologies and systems have been developed for the real-time and continuous monitoring of these components. Nevertheless, existing techniques and systems are currently optimized solely for the monitoring of the depth of anesthesia (DOA) and neuromuscular blockade, as outlined in current guidelines. These include the bispectral index (BIS^®^) for assessing DOA and neuromuscular monitoring (NM) for monitoring muscle function [1,2].

However, adequate direct monitoring of the third crucial factor of anesthesia has not been reliably established yet, that being adequate analgesia. For a long time, direct and validated monitoring of perioperative nociception was unavailable, and the anesthetist had to rely on indirect indicators such as blood pressure, heart rate, lacrimation, and sweating to estimate sufficient analgesia [3].

However, these surrogate parameters are not independent and can be susceptible to influences from various other factors, such as the type of surgery or anesthesia, blood loss, volume shifts, ambient temperature, or individual medications [4].

These associations can carry significant ramifications. Inadequate pain management may result in various complications, including elevated stress responses and postoperative pain due to suboptimal dosage. Conversely, the administration of high doses of medication may lead to an increased incidence of hemodynamic events, as well as postoperative nausea and vomiting [5].

In recent years, several monitoring systems have been developed to assess various parameters of nociception [6,7]. These techniques allow for the optimization of perioperative analgesic therapy and help to avoid overmedication.

The PMD-200-Nociception Monitor (NOL^®^; Medasense, Ramat Gan, Israel) is an approach that analyzes four different parameters (photoplethysmography, galvanic skin response, peripheral temperature, and accelerometery) related to the activation of the sympathetic nervous system non-invasively. These parameters are then integrated into an index known as the NOL^®^-Index, ranging from 0 to 100, with a target value of 10 to 25 [4,7].

Multiple studies have illustrated the efficacy of this system in detecting perioperative nociception [8,9]. However, whether this meaningfully impacts analgesic therapy has not been fully established [8,10]. Furthermore, NOL^®^ monitoring has the potential to impact postoperative opioid requirements and early postoperative pain, including its correlation with regional anesthesia [11].

Compared to traditional parameters such as heart rate and blood pressure, this method may offer advantages during general anesthesia. It can lead to reduced analgesia administration, prevents profound hemodynamic instability, and reduces postoperative pain [7,8,12].

Although the efficacy of this approach during general anesthesia has been validated through numerous studies, its value during epidural anesthesia remains inadequately investigated [12].

Epidural anesthesia is an established approach for supplementing general anesthesia and is associated with numerous benefits, such as reduced systemic opioid administration, diminished postoperative analgesic needs, and decreased occurrences of postoperative nausea and vomiting [13,14]. Nevertheless, there is no uniform implementation for the use of perioperative epidural anesthesia, whether for the choice of the drugs used or their optimal mode of administration (continuous vs. single dose) [15].

Therefore, this study aimed to conduct a retrospective analysis to assess the impact of perioperative epidural administration of local anesthetics during abdominal surgery on the NOL^®^-Index, while comparing it with established clinical parameters such as heart rate, blood pressure, and bispectral index. The effects were examined within the initial five minutes following epidural administration.

The investigation encompassed several additional facets. Specifically, it analyzed the ideal range of the NOL^®^-Index, delineated as falling between a score of 10 and 25, and the repercussions of administering epidurals within or below this threshold. Additionally, this study scrutinized the impacts of varying doses of epidural bupivacaine.

## 2. Materials and Methods

### 2.1. Ethical Considerations

This study was registered under the identifier DRKS00029120 on 1 July 2022, with the ethics committee approval (No. 2021-16201) obtained from the regional ethics committee of Rhineland Palatine, Mainz, chaired by Dr. Stephan Letzel. Informed consent was not deemed necessary.

### 2.2. Study Population

A retrospective analysis was conducted on 40 patients who underwent combined general and epidural anesthesia (thoracic levels T6–T12 with bupivacaine) for abdominal and urological surgeries between July 2022 and July 2023 at the University Medical Center of Johannes Gutenberg University in Mainz, Germany. The primary inclusion criterion for this study was the use of the NOL^®^-Index. Patients with conditions that could interfere with the NOL^®^ system, such as arrhythmia, severe hypothermia, cardiac pacemaker implantation, or high doses of vasoactive substances like norepinephrine, were not included. Consequently, only patients with an ASA status of 3 or lower were selected for this analysis. This study focused on the response of the NOL^®^ system following a single epidural administration of bupivacaine. Continuous infusions or the use of other local anesthetics were not considered in this study. Each administration was analyzed separately, even if multiple administrations occurred in a single patient. No patients were excluded from this study.

### 2.3. Study Design

All patients received extended hemodynamic monitoring (invasive blood pressure measurements, heart rate, oxygen saturation) and sedation monitoring utilizing the bispectral index monitoring system (BIS^®^ Monitoring System; Medtronic, Minneapolis, MN, USA). Nociception and stress levels were assessed using a dedicated monitoring system (NOL^®^; PMD-200; Medasense Biometrics Ltd., Ramat Gan, Israel). However, the anesthetist was not provided with guidance on interpreting the readings or adjusting the therapy accordingly. There was no difference between the patients in terms of the type of steering of the general anesthetic. The primary objective of this study was to assess and compare the impact of epidurally administered local anesthetics (bupivacaine 7.5 mg–25 mg) on the NOL^®^-Index, heart rate (HR), mean arterial blood pressure (MAP), and bispectral index (BIS). The epidural bupivacaine was administered as single or repeated injections rather than continuous infusions. The anesthetist did not follow a prescribed re-injection schedule based on regular intervals or specific stimuli. This study focused on the early primary phase immediately after the administration of bupivacaine. The NOL^®^-Index and the comparison parameters were analyzed at the time of administration (0 min) and during the following five minutes (1 min, 3 min, and 5 min).

A subanalysis evaluated the effect of different doses of bupivacaine on the NOL^®^-Index and the clinical parameters. For this purpose, a cut-off was defined at doses lower than 25 mg and at 25 mg. This criterion was based on the clinical practice at the authors’ hospital, where these doses of bupivacaine were most commonly used. However, applications with different doses were also included in this retrospective study and added to the low-dose group.

### 2.4. Data Splitting

The data were initially divided into different phases according to a specific protocol based on the defined thresholds of the NOL^®^-Index. This protocol classified values above 25 as indicative of a nociceptive event, values between 10 and 25 as an optimal nociceptive range, and values below 10 as indicative of the absence of nociceptive activation.

In the initial phase, all recorded instances of epidural bupivacaine administrations (*n* = 119) were reviewed. Following this, the investigation homed in on instances where the NOL^®^-Index at the time of application exceeded a minimum threshold of 10 (*n* = 89), and, subsequently, a threshold of at least 25 (*n* = 35). During the third phase, the total number of administered epidural doses was categorized into the following two groups: a lower 25 mg (12.2 ± 2.7 mg) bolus group (*n* = 73) and a 25 mg bolus group (*n* = 46). Within each group, subgroups were formed with a NOL^®^-Index of at least 10 and at least 25 (lower 25 mg: NOL^®^ > 10 *n* = 55, NOL^®^ > 25 *n* = 22; 25 mg: NOL^®^ > 10 *n* = 34, NOL^®^ > 25 *n* = 13) (Figure 1).

The secondary objective involved comparing the temporal progression of the NOL^®^-Index among the various doses of epidural bupivacaine (Figure 1).

### 2.5. Statistical Analyses

Statistical analyses were performed using GraphPad Prism 9 software (GraphPad Software, Boston, MA, USA). Given the pilot nature of the analysis, no pre-data collection or statistical planning occurred, and the presented analyses are primarily descriptive.

The repeated measures underwent one-way ANOVA, with time-point comparisons being performed utilizing the Holm–Sidak test. *p*-values less than 0.05 were considered statistically significant. For better comparability of the different NOL^®^-Index values in the figures, these are given as percentages. All figures depict boxplots featuring the median and quartiles. The plot illustrates the interquartile range and demonstrates the percentage relative to the baseline. In order to compare the various doses, a one-sample t-test and a Wilcoxon test were conducted.

## 3. Results

### 3.1. Included Measurements

This retrospective data analysis included a cohort of 40 ASA II and ASA III patients, encompassing 119 instances of single or multiple administrations of bupivacaine during abdominal and urological surgeries. The measurement was divided up as described above. The populations and the number of different operations are summarized in Table 1. The raw data can be found in the Appendix A (Appendix A).

### 3.2. Comparative Analysis

It was noted that the NOL^®^-Index exhibited a significant decrease within the assessed timeframe following the administration the epidural analgesics. Conversely, conventional clinical parameters displayed no discernible changes and failed to mirror the observed effect. Notably, this effect was particularly pronounced when the NOL^®^-Index surpassed the threshold of 25 (Figure 2).

However, the impact of antinociceptive therapy was also evident when this parameter exceeded 10 (Figure 3) or when the NOL^®^-Index was less than 10 (Table 2.).

### 3.3. Dose Analysis

The analysis of varying epidural doses demonstrated a marked decrease in nociception signals within the high-dose group (administered with 25 mg of bupivacaine). This effect was particularly notable in the subgroup adhering to a threshold of 25, exhibiting statistical significance (Figure 4). During the initial five minutes following epidural administration, no significant hemodynamic response was observed in any group or in the BIS^®^ (Table 1 and Figure 4).

## 4. Discussion

This retrospective study presents novel findings indicating the NOL^®^-Index as a highly efficacious method for the illustration of the antinociceptive impact of epidural local anesthetics. The NOL^®^-Index exhibited in this context a notably superior ability to visualize this effect compared to conventional parameters such as heart rate, mean arterial blood pressure, or bispectral index (BIS^®^), which did not exhibit significant effects.

Since this study examined the effect of epidural bupivacaine on the NOL^®^-Index retrospectively, a direct intervention was not possible. Consequently, the NOL^®^-Index did not influence the administration of anesthesia. As a result, regional anesthesia was administered even when the NOL^®^-Index was below 25 or even 10. This is essential because the NOL^®^ system defines a value above 25 as indicative of a nociceptive event, and a value between 10 and 25 as an optimal nociceptive range [4,7]. Therefore, the authors decided to analyze the data in three groups, as follows: one group included all doses, one group included only those with values above 10, and the third group included only those with values above 25 [4,7].

The significant decrease was especially pronounced if the anesthetics were administered when the NOL^®^-Index was over the value of 25 [16]. This is congruent with the recommended range of between 10 and 25, where values above 25 are associated with high stress and nociception [17]. In previous studies, the NOL^®^-Index has consistently shown a superior performance compared to HR and MAP in distinguishing between noxious and non-noxious stimuli, as evidenced in various events such as intubation or skin incision [18,19]. For patients undergoing Video-Assisted Thoracoscopic Surgery (VATS), comparable results have been documented, with NOL^®^ demonstrating greater efficacy in evaluating different events [20]. This study is particularly noteworthy as it showcases the effectiveness of the NOL^®^ system when used in conjunction with epidural anesthesia, suggesting a potentially valuable combination [20].

The use of the NOL^®^-Index in this context could provide clarity and potentially improve patient safety by facilitating the development of an optimized approach. During daily clinical practice, there is considerable heterogeneity in the management of perioperative epidural analgesia, with no standardized guidelines or instructions for optimal use. Diverse perspectives exist regarding the selection of the initial drug, its dosage, and adequate co-analgesics, with no standardized guidelines or recommendations available. The literature remains inconclusive regarding optimal practices in this regard [21].

Numerous studies have shown the potential of the NOL^®^-Index for this purpose. Investigation of this context was notably focused within the domain of general anesthesia. For instance, reducing NOL^®^ usage has led to decreased perioperative opioid consumption and shortened extubation times [22]. The opioid-sparing effect can be characterized by a reduction of 22% [23]. Although the opioid-sparing effect was measurable in most instances, there were studies in which it was not observed [10]. Conversely, in these trials, NOL^®^ demonstrated a positive effect on postoperative pain outcomes [5].

These findings suggest the potential applicability of NOL^®^ for guiding analgesic therapy and its potential transferability to the context of epidural anesthesia.

The secondary analysis of this study elucidates the impact of varying epidural doses of bupivacaine on the NOL^®^-Index and other parameters included in this study. This stratification is based on clinical practice, where the epidural administration of bupivacaine at doses of 12.5 mg or 25 mg is commonly observed among clinicians. In this context, the NOL^®^-Index demonstrated a significantly faster decrease in the high-dose group.

Comparable effects were observed following the assessment of standardized nociceptive tetanic stimuli with varying doses of remifentanil. The NOL^®^-Index proved effective in discerning this effect [24]. This particular feature of the NOL^®^-Index can have a significant impact on the efficacy of analgesic therapy, as it can help to prevent both underdosing and overdosing [5].

This study possesses several limitations, with the primary concern being its retrospective nature. Additionally, it lacks involvement in a sizable prospective randomized trial featuring distinct groups, where the integration of the NOL^®^-Index is incorporated as an intervention within a designated group. Therefore, this study was not specifically designed to yield these outcomes with robust statistical power. This is especially crucial for analyzing the epidural dosage, given the limitation of the relatively small number of analyzed applications. However, the noteworthy level of significance observed stresses the informative quality of the results.

Another limitation of this study is the inclusion of different surgical procedures, each with varying levels of surgical complexity and nociceptive stress. This variation could serve as a confounding factor, making it ideal to compare procedures within the same surgical category. However, it is important to emphasize that this study specifically focuses on the effects of epidural bupivacaine application on the various parameters, rather than on comparing different surgical operations.

An added constraint of this study design is the evaluation of the NOL^®^-Index only within the subsequent five minutes following analgesic administration. This time frame was chosen because our initial clinical pilot analysis indicated that the primary effect occurs directly after application. Longer time frames introduced significant interference due to the retrospective nature of this study and intraoperative events. This interference explains the large error bars in the graphs, which could be minimized by defining a shorter time frame.

This issue is particularly relevant in the context of the single-shot administration of bupivacaine. The literature reports varying onset times, typically around 15 to 20 min, especially for epidural dosages. However, it is important to note that these studies measured the time from administration to the achievement of full sensory or motor block [25,26]. Hemodynamic effects manifest with a longer delay than sensory effects, which may account for the findings observed in our study [27]. The effect on nociception, measurable with nociception monitoring systems, has not been the primary focus of these studies. The results of this study could suggest that the onset time of bupivacaine is much faster for nociceptive effects due to the rapid action on Aδ and C- fibers, which are crucial for nociception and temperature sensation because of their extremely thin myelination [26]. This also explains why no effects on physiological parameters were detected. Longer time frames in future studies will be necessary to address these effects more thoroughly.

Another limitation of this study is the fact that nociceptive stimuli, upon cerebral processing, lead to a sympathetic stress reaction. This sympathetic response is notably attenuated with an epidural catheter, which directly influences the NOL^®^-Index. This effect is also discernible in established clinical parameters such as heart rate and arterial pressure [28]. For optimized evaluation, a comparison with more precise methods, such as direct cerebral assessment of nociceptive stimuli or the direct hormone activity of stress, would be necessary [29].

## 5. Conclusions

In conclusion, this retrospective study represents the first demonstration of the effectiveness of the NOL^®^-Index in measuring nociceptive effects following epidural administration of bupivacaine within the time frame of the initial few minutes, showing potential superiority over conventional clinical parameters. Furthermore, this study has demonstrated the sensitivity of the NOL^®^-Index in distinguishing between different dose effects and its high sensitivity in detecting the analgesic effect of bupivacaine.

## Figures and Tables

**Figure 1 jcm-13-04968-f001:**
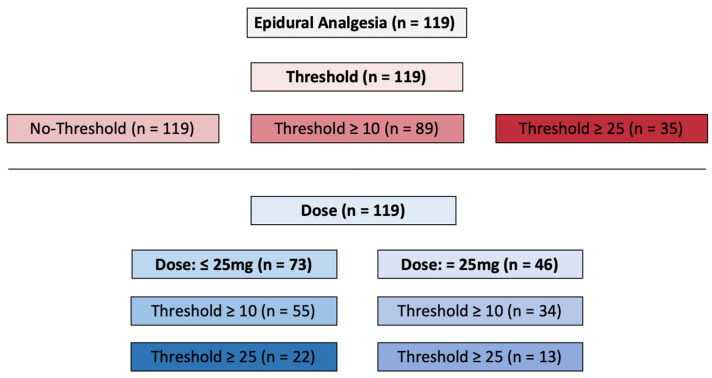
The applications of different epidural analgesics are divided into different groups based on their NOL^®^—threshold and/or dosage.

**Figure 2 jcm-13-04968-f002:**
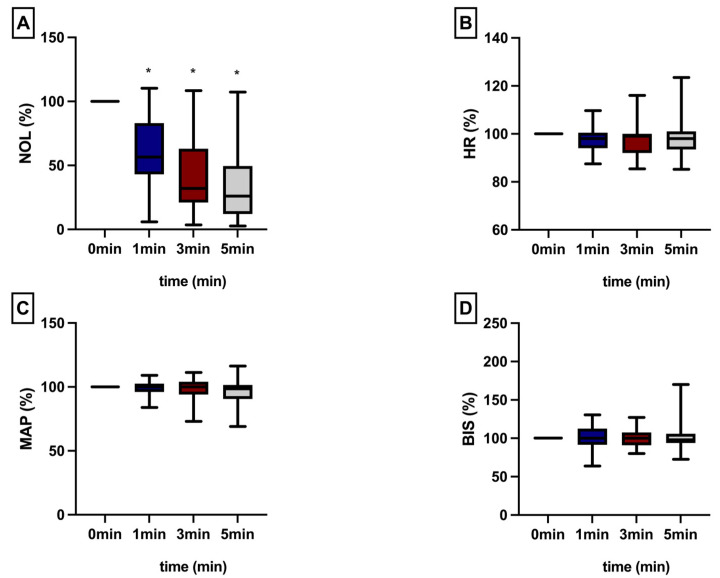
The assessment of all parameters (NOL = NOL^®^-Index (**A**), Heart Rate = HR (**B**), Mean Arterial Pressure = MAP (**C**), Bispectral Index^®^ = BIS (**D**)) with a NOL^®^-threshold greater than or equal to 25 was performed immediately and at 1, 3, and 5 min following epidural administration. All values are presented as a percentage of the baseline. * intragroup significant *p* ≤ 0.05 (one-way ANOVA with time-point comparisons were performed utilizing the Holm–Sidak test).

**Figure 3 jcm-13-04968-f003:**
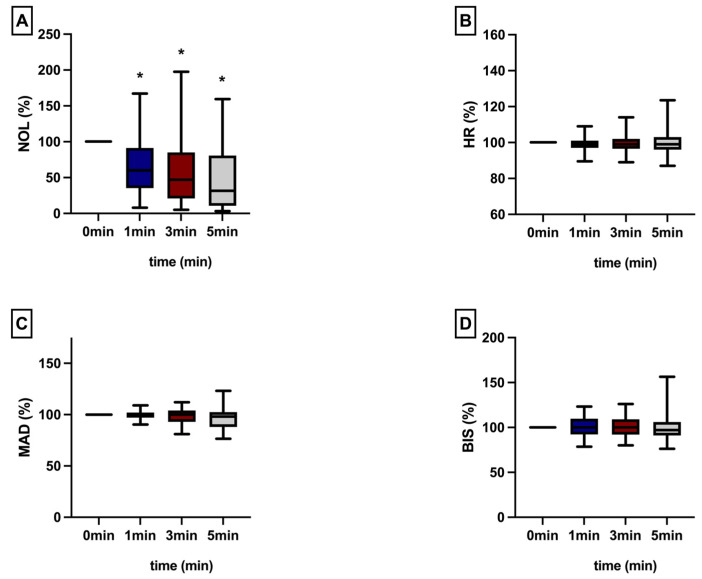
The assessment of all parameters (NOL = NOL^®^-Index (**A**), Heart Rate = HR (**B**), Mean Arterial Pressure = MAP (**C**), Bispectral index = BIS^®^ (**D**)) with a NOL^®^-threshold greater than or equal to 10 was performed immediately and at 1, 3, and 5 min following epidural administration. All values are presented as a percentage of the baseline. * intragroup significant *p* ≤ 0.05 (one-way ANOVA with time-point comparisons were performed utilizing the Holm–Sidak test).

**Figure 4 jcm-13-04968-f004:**
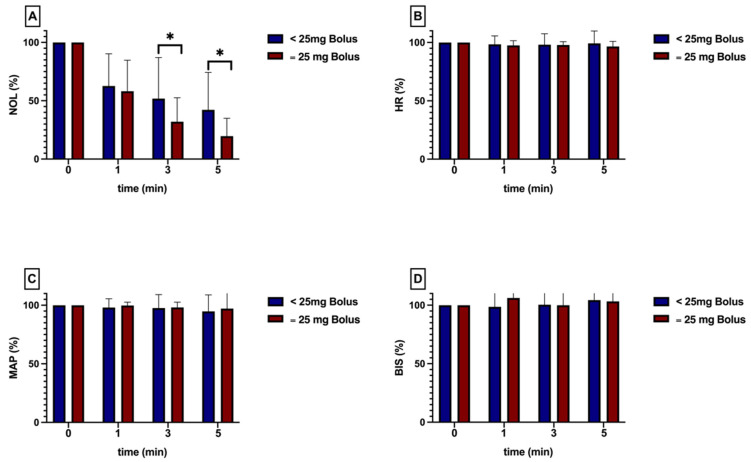
The NOL^®^-Index (**A**) was compared following the administration of a bolus dose equal to 25 mg versus a dose less than 25 mg with a NOL^®^-threshold equal to 25. At one, three, and five minutes post administration, the NOL^®^-Index was significantly lower in the 25 mg group. A similar effect was not measurable in the conventional parameters (HR (**B**), MAP (**C**), and BIS^®^ (**D**)). All values are presented as a percentage of the baseline. * intragroup significant *p* ≤ 0.05 (one-sample *t*-test and Wilcoxon test).

**Table 1 jcm-13-04968-t001:** Population and procedures. An overview of the included patient population (age, sex, and anesthetic status) and the procedures performed.

Population	
Age	63.6 ± 13.1
Gender	28 male
	12 female
Status	20 ASA II
	20 ASA III
Procedures	
Esophagectomy	14
Pancreatectomy	11
Nephrectomy	4
Colectectomy	4
Sarcoma resection	2
Herniotomy	2
Liver resections	1
Gastrectomy	1
Cystectomy	1

**Table 2 jcm-13-04968-t002:** Evaluation of all parameters over five minutes.

	Parameter	0 min	1 min	3 min	5 min
Bupivacaine Total *n* = 119	NOL	19 ± 14	14 ± 11	13 ± 11 *	10 ± 10 *
	MAP	86 ± 14	85 ± 14	85 ± 12	83 ± 13
	HR	68 ± 14	68 ± 14	68 ± 13	68 ± 13
	BIS	42 ± 10	42 ± 10	42 ± 10	42 ± 10
Bupivacaine Total ≥ 10 *n* = 89	NOL	25 ± 13	16 ± 12 *	15 ± 10 *	11 ± 9 *
	MAP	86 ± 13	85 ± 14	84 ± 11	82 ± 12
	HR	68 ± 11	67 ± 11	68 ± 11	68 ± 10
	BIS	43 ± 10	42 ± 10	43 ± 9	42 ± 9
Bupivacaine Total ≥ 25 *n* = 35	NOL	38 ± 11	22 ± 13 *	17 ± 11 *	12 ± 10 *
	MAP	87 ± 12	86 ± 13	85 ± 13	84 ± 13
	HR	69 ± 12	68 ± 12	67 ± 11	68 ± 11
	BIS	41 ± 9	41 ± 9	40 ± 8	40 ± 7
Bupivacaine < 25 mg *n* = 73	NOL	20 ± 14	14 ± 11	13 ± 10 *	12 ± 10 *
	MAP	85 ± 14	84 ± 14	84 ± 12	82 ± 12
	HR	67 ± 11	67 ± 11	67 ± 11	67 ± 11
	BIS	43 ± 10	43 ± 11	43 ± 10	42 ± 10
Bupivacaine < 25 mg NOL ≥ 10 *n* = 55	NOL	26 ± 13	17 ± 12 *	15 ± 10 *	12 ± 9 *
	MAP	87 ± 13	86 ± 13	84 ± 11	82 ± 11
	HR	68 ± 11	68 ± 11	68 ± 11	68 ± 10
	BIS	43 ± 10	42 ± 10	43 ± 9	42 ± 9
Bupivacaine < 25 mg NOL ≥ 25 *n* = 22	NOL	38 ± 11	23 ± 12 *	19 ± 11 *	15 ± 10 *
	MAP	83 ± 10	82 ± 11	77 ± 21	80 ± 13
	HR	67 ± 10	66 ± 11	65 ± 10	66 ± 10
	BIS	42 ± 10	42 ± 11	41 ± 9	41 ± 8
Bupivacaine = 25 mg *n* = 46	NOL	19 ± 14	14 ± 11	12 ± 12 *	9 ± 9 *
	MAP	87 ± 15	86 ± 15	86 ± 13	85 ± 14
	HR	71 ± 17	70 ± 17	70 ± 16	70 ± 16
	BIS	40 ± 10	41 ± 9	40 ± 10	41 ± 10
Bupivacaine = 25 mg NOL ≥ 10 *n* = 34	NOL	24 ± 12	16 ± 12 *	12 ± 13*	9 ± 10 *
	MAP	89 ± 14	88 ± 14	88 ± 12	87 ± 14
	HR	71 ± 17	70 ± 16	70 ± 15	71 ± 16
	BIS	40 ± 10	40 ± 9	39 ± 10	40 ± 11
Bupivacaine = 25 mg NOL ≥ 25 *n* = 13	NOL	38 ± 10	22 ± 13	13 ± 10 *	8 ± 8 *
	MAP	94 ± 12	94 ± 13	92 ± 11	90 ± 11
	HR	73 ± 14	71 ± 13	71 ± 13	70 ± 12
	BIS	38 ± 5	40 ± 4	38 ± 6	39 ± 4

Evaluation of all parameters (NOL = NOL^®^-Index, Heart Rate = HR, Mean Arterial Pressure = MAP, Bispectral index = BIS^®^) was conducted immediately and at 1, 3, and 5 min following epidural administration. NOL^®^ and BIS^®^ are expressed as index values, HR is presented in beats per minute (min^−1^), and MAP is given in mmHg. * intragroup significant *p* ≤ 0.05 (one-way ANOVA with time-point comparisons were performed utilizing the Holm–Sidak test).

## Data Availability

The original contributions presented in this study are included in the article. Further inquiries (raw data) can be directed to the corresponding author.

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
