# Peer review of "Analysis of the Effects of Epidural Anesthesia on the Nociception Level Index (NOL®) during Abdominal Surgery"

_jcm, 2024, doi:10.3390/jcm13164968_

Round 1

Reviewer 1 Report

Comments and Suggestions for Authors

In this manuscript authors present the use of NOL system in epidural analgesia during abdominal surgery. 

Methods:

1. patients - there is no clear explanation on patient selection (inclusion and exclusion criteria) and there is no basic information about patients, i.e. demographic data.

2. intervention - although this was not an interventional study, authors should better describe the application of local anesthetic. Was there a protocol by which the local anesthetic was administered? The dosage is unclear (bupivacaine 7.5-25 mg), does it refer to single or repeated boluses, is it cumulative dose? When were these boluses administered, at regular intervals or according to stimuli (NOL index, physiologic reactions...)? Were there patients with continuous infusions of local anesthetics administered epiduraly?

3. data splitting section is unclear, as is Figure 1. Consider using flowchart for this purpose as it would make data more clear. Why were patients categorized in two groups, lower than 25 mg and 25 mg bolus, why did authors chose 25 mg as a marker? What were median bolus doses in these groups? 

4. Why was period of 5 minutes following epidural administration used, specially when it is known that bupivacaine has longer time of onset? I suppose physiological parameters were measured at same time points? It says in results section, 3.2. Dose analysis, that "during initial 5 minutes following peidural administration, no significant hemodynamic response was observed...". Was there any hemodynamic response after that period, specially in patients receiving higher boluses?

5. Figures 2 and 3 - graphs are labeled incorrectly. It says in the legend that MAP is shown in graph B, and that HF is shown on graph C. Furthermore, all abbreviations in the figures should be explained in the Figure legends. Please explain the unit of measure on vertical axis, what is % reffered to?

6. Table - please add table title above the table. Please add explanations of abbreviations in the table legend. 

7. Figure 7. In the vertical axis, why is NOL presented as %? As percentage of what? Why didn't authors use absolute NOL values? 

Author Response

Dear Reviewer,

Thank you for the valuable and constructive comments on the presented study. Relevant changes in the manuscript have been highlighted. The introduction has been updated, particularly the materials and methods section, and the discussion section has also been revised.

  1. patients - there is no clear explanation on patient selection (inclusion and exclusion criteria) and there is no basic information about patients, i.e. demographic data.
  • Thank you for this suggestion. We have added the information about patient selection, population characteristics, and the different procedures to the Methods and Results sections (Table 1). Patients with primary conditions that interfere with the use of the NOL® system, such as arrhythmia, extreme hypothermia, caridac pacemaker, or a high rate of vasoactive substances like norepinephrine, were excluded from this study. Consequently, only patients with an ASA status of 3 or lower were included in this investigation.
  1. intervention - although this was not an interventional study, authors should better describe the application of local anesthetic. Was there a protocol by which the local anesthetic was administered? The dosage is unclear (bupivacaine 7.5-25 mg), does it refer to single or repeated boluses, is it cumulative dose? When were these boluses administered, at regular intervals or according to stimuli (NOL index, physiologic reactions...)? Were there patients with continuous infusions of local anesthetics administered epiduraly?
  • Thank you for this question. The epidural local anesthetics were administered as single or repeated injections rather than continuous infusions. The anesthetist did not follow a prescribed re-injection schedule based on regular intervals or specific stimuli. This information were added to method section.
  1. data splitting section is unclear, as is Figure 1. Consider using flowchart for this purpose as it would make data more clear. Why were patients categorized in two groups, lower than 25 mg and 25 mg bolus, why did authors chose 25 mg as a marker? What were median bolus doses in these groups? 
  • Thank you for your question. We have clarified the data splitting in the Methods and Discussion sections. The data were initially divided into different phases based on the thresholds of the NOL-Index®. Values above 25 were classified as indicative of a nociceptive event, values between 10 and 25 as an optimal nociceptive range, and values below 10 as indicative of the absence of nociception.
  • The bupivacaine dosage groups were defined based on clinical practice, where the epidural administration of bupivacaine at doses of 12.5 mg or 25 mg is commonly observed among different clinicians. The mean of the lower bupivacaine dose was included in the manuscript. Figure 1. was modified.
  1. Why was period of 5 minutes following epidural administration used, specially when it is known that bupivacaine has longer time of onset? I suppose physiological parameters were measured at same time points? It says in results section, 3.2. Dose analysis, that "during initial 5 minutes following peidural administration, no significant hemodynamic response was observed...". Was there any hemodynamic response after that period, specially in patients receiving higher boluses?
  • Thank you for this question. This time frame was chosen based on our initial clinical pilot analysis, which indicated that the primary effect occurs immediately after application. Physiological parameters were measured at the same time points but showed no effect even in the high-dose group. We appreciate your input and will discuss this issue in greater detail in the discussion section.
  1. Figures 2 and 3 - graphs are labeled incorrectly. It says in the legend that MAP is shown in graph B, and that HF is shown on graph C. Furthermore, all abbreviations in the figures should be explained in the Figure legends. Please explain the unit of measure on vertical axis, what is % reffered to?
  • Thank you for this note. The graph labels have been corrected, and all abbreviations in the figure have been explained. All values are presented as a percentage of the baseline.
  1. Table - please add table title above the table. Please add explanations of abbreviations in the table legend. 
  • Thank you for this note. A title and explanations of the abbreviations were explained.
  1. Figure 7. In the vertical axis, why is NOL presented as %? As percentage of what? Why didn't authors use absolute NOL values
  • Thank you for this suggestion. All values are presented as a percentage of the baseline. The decision to present the data as percentages is based on the fact that it makes visual comparison of the values easier to understand. This does not affect the significance.

Reviewer 2 Report

Comments and Suggestions for Authors

Thank you for the opportunity to provide a review of the article.

My rating of the paper is very high; however, below are some comments that you should consider.

 1. in the Abstract, you should state the type of procedure for which the epidural catheter was placed. Patient information and types of procedures should be included in this section. The wording “thoracic epidural catheter administration” alone is not sufficient.

2. in the Abstract in the Results section, please include the primary results, not the mere description “This study showed a significant decrease in NOL.”

3. line 48: please include other factors that affect perioperative nociception

4. Line 52: I suggest instead of excessive medication to use another wording indicating the administration of specific groups of drugs

5. line 95: please explain the differences in the dosage of bupivacaine and the potential impact on the results obtained

6. please provide a more precise dosing schedule for bupivacaine epidural. I don't understand the sentence Line 122-123. under what circumstances was decided “119 instances of local anesthetic administration during abdominal surgery” in a group of 40 patients. The method of administration and scheme needs to be discussed in more detail

7 Conclusions need re-examination. Shouldn't we use :measuring instead of the phrase assessing nociceptive effects.

8 The phrase within the first few minutes is unclear to me. Do the conclusions refer only to those first few minutes after administration of the drug, or the entire period of action of epidural administrations of bupivacaine.

9. lines 221-226 need to be improved and completed.

Author Response

Thank you for the valuable and constructive comments on the presented study. Relevant changes in the manuscript have been highlighted. The abstract and the introduction has been updated, particularly the materials and methods section, and the discussion section has also been revised.

  1. in the Abstract, you should state the type of procedure for which the epidural catheter was placed. Patient information and types of procedures should be included in this section. The wording “thoracic epidural catheter administration” alone is not sufficient.
  • Thank you for this suggestion. The epidural catheter was used during abdominal and urological surgeries. Detailed patient information has been added to the manuscript as Table 1.
  1. in the Abstract in the Results section, please include the primary results, not the mere description “This study showed a significant decrease in NOL.”
  • Thank you for this suggestion. The primary results have been included in this section.3. line 48: please include other factors that affect perioperative nociception
  1. line 48: please include other factors that affect perioperative nociception
  • Several other factors that affect perioperative nociception, such as type of surgery and anaesthesia, were included.
  1. Line 52: I suggest instead of excessive medication to use another wording indicating the administration of specific groups of drugs
  • Thank you for this suggestin, the sentence was change to: “Conversely, the administration of high doses of medication may lead to an increased incidence of hemodynamic events as well as postoperative nausea and vomiting.”
  1. line 95: please explain the differences in the dosage of bupivacaine and the potential impact on the results obtained
  • Thank you for the suggestion. The concept of dose analysis for bupivacaine is explained in more detail in Section 2.3, Study Design.
  1. please provide a more precise dosing schedule for bupivacaine epidural. I don't understand the sentence Line 122-123. under what circumstances was decided “119 instances of local anesthetic administration during abdominal surgery” in a group of 40 patients. The method of administration and scheme needs to be discussed in more detail
  • Thank you for this suggestion. This section has been discussed in more detail, and the schematic representation has been updated.
  1. Conclusions need re-examination. Shouldn't we use :measuring instead of the phrase assessing nociceptive effects.
  • Thank you for your comment. The term "assessing" has been replaced with "measuring."
  1. The phrase within the first few minutes is unclear to me. Do the conclusions refer only to those first few minutes after administration of the drug, or the entire period of action of epidural administrations of bupivacaine.
  • Thank you for this suggestion. The conclusion has been revised to clarify that only the time frame of the first five minutes was analyzed.
  1. lines 221-226 need to be improved and completed.
  • Thank for this hint. The Institutional Review Board Statement was completed.

Round 2

Reviewer 1 Report

Comments and Suggestions for Authors

The authors responded to all questions and made changes to the manuscript. There are, however, few more issues I would like the authors to address:

1. Patients: How many patients underwent described procedure (general+epidural aensthesia) for abdominal and urological surgery during study period? Were there patients excluded from study and why? This data should be presented as a flowchart to make data more clear.

2. It is clear that patients underwent different procedures, some of which are associated with more or less pain than others. This bias makes data interpretation difficult and could be a confounding factor. It would be better if type of the surgery is more uniform, but in this case this should be mentioned as a limitation at least.

3. Tables: Name of the table should be presented above the table, not in the legend. 

4. Figure 2. It says in the legend that values are presented as a percentage of the baseline. However, in the graphs there are clearly stated units of measure (i.e. mmHg).

5. In table 2 authors should consider stating how many measurements were performed for each of the categories. For example, Bupivacain total (n= number)

Author Response

Thank you for the suggestion. All questions have been addressed, and all changes have been highlighted in yellow.

  1. Patients: How many patients underwent described procedure (general+epidural aensthesia) for abdominal and urological surgery during study period? Were there patients excluded from study and why? This data should be presented as a flowchart to make data more clear.
  • Thank you for the suggestion. The authors agree that a flow chart would be highly beneficial. Theoretically, the number of patients undergoing combined procedures was significantly higher during this timeframe. The primary inclusion criterion for this retrospective study was the mandatory use of the NOL-index. The NOL-index was utilized according to the manufacturer's guidelines, exclusively in patients without arrhythmias, hypotension, or high levels of vasoactive substances. No patients who received the combination of general anesthesia, epidural anesthesia, and NOL monitoring were excluded from this study. This clarification is included in the Methods section.
  1. It is clear that patients underwent different procedures, some of which are associated with more or less pain than others. This bias makes data interpretation difficult and could be a confounding factor. It would be better if type of the surgery is more uniform, but in this case this should be mentioned as a limitation at least.
  • Thank you for the note. This is an important point, and it has been added to the limitations section.
  1. Tables: Name of the table should be presented above the table, not in the legend. 
  • Thank you for the suggestion. The table titles have now been placed above the respective tables
  1. Figure 2. It says in the legend that values are presented as a percentage of the baseline. However, in the graphs there are clearly stated units of measure (i.e. mmHg).
  • Thank you for the suggestion. The errors have been corrected in Figure 2, and the units have been specified as percentages.
  1. In table 2 authors should consider stating how many measurements were performed for each of the categories. For example, Bupivacain total (n=number)
  • Thank you for the suggestion. The number of measurements has been included in Table 2.